# SAC: Adaptive Learning Rate Scaling with Architectural Constraints

## Abstract

The design of optimizers for modern Large Language Models (LLMs) is governed by the critical trade-off between performance, memory footprint, and computational throughput. High-accuracy methods, such as those exploiting gradient preconditioning techniques, are often memory-intensive and may introduce significant computational overhead, while efficient ones like Galore may not reach the same performance level. In this work, we present **S**caling with **A**rchitectural **C**onstraints (**SAC**), an optimizer wrapper that navigates these competing demands for the first time. SAC enhances existing adaptive optimizers by modulating per-parameter learning rates with lightweight, hierarchical constraints derived from model architectures. On the C4 pre-training benchmark, SAC+AdamW achieves state-of-the-art perplexity from 60M to 3B model sizes, converging faster without incurring the high costs of complex preconditioning. It also enhances training stability, showcasing robustness across varied learning rates and batch sizes. Qualitatively, empirical analysis shows that SAC fosters a more coordinated optimization process, leading to improved gradient dynamics. Its versatility has been further validated by the strong results across downstream tasks and domains, including long sequence modeling, parameter-efficient fine-tuning, image classification with diverse models like ViTs and CNNs, and evaluations on multimodal benchmarks.

## 1 Introduction

Optimizing large-scale networks like Large Language Models (LLMs) (Liu et al., 2024a; Achiam et al., 2023) remains a core challenge in modern machine learning. As models grow from millions to billions of parameters, the gap between architectural complexity and optimizer simplicity widens. Most training algorithms can be viewed through two orthogonal design principles: **(i) Temporal smoothing** refers to using historical gradient moments (*e.g.*, momentum, variance estimates) to stabilize the optimization trajectory over time. Techniques like Adam (Kingma & Ba, 2015) exploit this principle to reduce oscillations and help convergence. **(ii) Spatial Structuring** refers to applying constraints, preconditioning, or coordination across parameters that share architectural relationships (*e.g.*, within the same block or layer), rather than treating each weight independently.

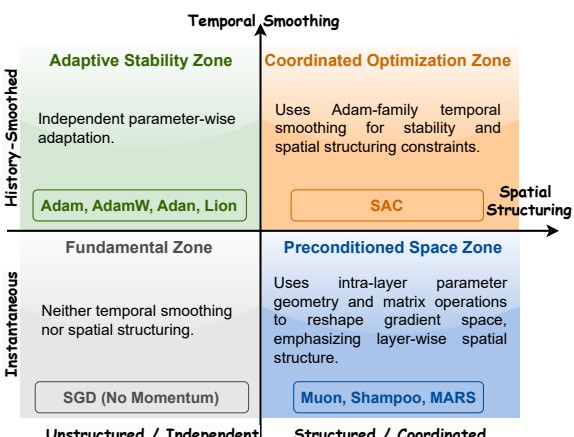

Figure 1: **The Optimizer Design Plane**. We map the landscape of optimizers along two key axes: **Temporal Smoothing** (Y-axis), which describes how much an optimizer relies on historical gradients for stability, and **Spatial Structuring** (X-axis), which reflects the use of architectural hierarchy to coordinate model updates.

The interplay of these principles gives rise to a design landscape as shown in Figure 1. Most widely used optimizers, including Adam (Kingma & Ba, 2015), AdamW (Loshchilov & Hutter, 2019), Lion (Chen et al., 2023), and Adan (Xie et al., 2023), occupy the Adaptive Stability Zone. Their strength is exceptional temporal smoothing, which provides stable updates and robust parameter-

wise adaptation. Yet they remain agnostic to the model's architecture, leading to uncoordinated updates between attention heads and feed-forward layers. Even memory-efficient variants like Adafactor (Shazeer & Stern, 2018), Adam-mini (Zhang et al., 2024), CAME (Luo et al., 2023), and APOLLO (Zhu et al., 2024a) cannot remedy this structural blindness; they simply reduce storage while maintaining per-parameter independence.

At the opposite extreme lies the Preconditioned Space Zone, populated by second-order or geometric optimizers such as Shampoo (Gupta et al., 2018), Muon (Jordan et al.), and MARS (Yuan et al., 2024). These methods reshape the gradient space using matrix operations to align updates coherently within layers or tensors. While theoretically elegant, they incur substantial computational and memory costs. Furthermore, their focus is often limited to local, layer-wise structure and typically sacrifices the robust temporal smoothing that makes Adam variants practical for large-scale training.

This leaves a critical quadrant unoccupied: the Coordinated Optimization Zone, which promises the stability of temporal smoothing combined with the intelligence of spatial structuring. The structural blindness of Adam is increasingly linked to training instabilities in deep models, such as optimization discrepancies between shallow and deep layers and sudden loss spikes (Molybog et al., 2023). As hardware acceleration (Dao, 2023) and distributed frameworks (Shoeybi et al., 2019) make full-parameter training more feasible, the primary bottleneck is shifting from raw efficiency to the effective coordination of parameter updates. In this work, we introduce **S**caling with **A**rchitectural **C**onstraints (**SAC**), an optimizer wrapper designed to bridge this gap and inhabit the Coordinated Optimization Zone. SAC's core idea is to modulate the per-parameter adaptive learning rates of an Adam-family optimizer with lightweight, hierarchically-derived constraints from the model's architecture. It retains the proven temporal stability of Adam while introducing intelligent spatial coordination, with negligible computational overhead. Empirically, we find that both layer-wise homogenization and block-wise heterogenization constraints could be beneficial for LLM training, which aligns with previous findings (Molybog et al., 2023; Zhang et al., 2025).

To rigorously validate the effectiveness of SAC, we conduct extensive experiments across a comprehensive suite of tasks and models, including C4 pre-training, supervised fine-tuning (SFT) on GLUE benchmark, parameter-efficient fine-tuning (PEFT) on commonsense reasoning tasks, and multiple MLLM and vision benchmarks. In addition, to demonstrate its broader applicability, we extend our evaluation to computer vision tasks, including classical image classification on CIFAR and ImageNet. The results demonstrate that SAC consistently outperforms baseline optimizers and relevant optimizers, achieves faster convergence, improves model performance, and exhibits robust stability across various settings – yielding up to 30% improvements over baselines. These findings demonstrate the substantial benefits and untapped potential of architectural constraints for LLM optimization.

Our contributions can thus be summarized as follows:

- We identify and analyze two key orthogonal design principles, i.e., **temporal smoothing** and **spatial structuring**, to systematically classify optimization algorithms. This framework reveals a critical gap in the current landscape: the absence of a practical method that effectively combines the stability of historical gradient smoothing with the intelligence of architectural coordination.

- To fill this gap, we propose SAC, an optimizer wrapper that pioneers the Coordinated Optimization Zone. Implemented as a versatile wrapper, SAC seamlessly integrates with existing adaptive optimizers and PEFT techniques, requiring no changes to model architectures and incurring negligible overhead. We also provide CPU, GPU, and hybrid implementations to accommodate different computational trade-offs for practical deployment.

- The consistent superiority of SAC across tasks and model sizes suggests the potential of learning rate scaling with architectural priors. We suppose that it could still be beneficial in training larger-scale models, and hope it can inspire further exploration in the community along this line.

## 2 METHODOLOGY

As established, the design of modern optimizers can be deconstructed into two orthogonal principles: temporal smoothing and spatial structuring. Canonical adaptive optimizers compute parameter-wise adaptive learning rates $\alpha_t \in \mathbb{R}^{m \times n}$ that impose *historical gradient constraints* on the optimization

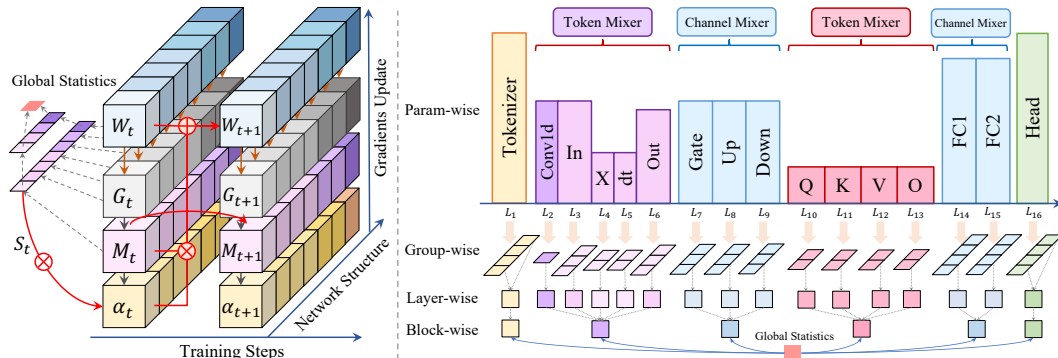

Figure 2: **The SAC Optimization Pipeline**. **(a) Left**: SAC is wrapped upon an adaptive learning rate optimizer. At each step $t$, the base adaptive optimizer controls the update of gradients $G_t$ to model weights $W_{t+1}$ by computing parameter-wise learning rates $\alpha_t$ with historical constraints to modulate the first-order moments $M_t$. SAC introduces parallel scale factors $S_t$ with architectural constraints. **(b) Right**: The scale factors estimation pipeline. Taking the hybrid model with a Mamba block (Gu & Dao, 2023) and a self-attention block as an example, the model can be partitioned into four types of blocks (Yu et al., 2024), *i.e.*, Tokenizer, Token Mixer, Channel Mixer, and Output Head. Scale factors are then applied at multiple granularities: parameter-wise, group-wise, layer-wise, and block-wise.

trajectory over time. However, they lack the latter. SAC systematically unifies both principles by introducing a structured, coarse-grained scale factor $S_t$ that imposes spatial architectural constraints as shown in Figure 2. Our SAC braids these signals into a hierarchical weight update rule:

$$W_{t+1} = W_t - \underbrace{\eta \cdot \alpha_t \cdot M_t}_{\text{Temporally-Smoothed}} \cdot \overbrace{S_t}^{\text{Spatial Structured Scale Factor}} \tag{1}$$

where $(\eta \cdot \alpha_t \cdot M_t)$ is the standard update from a base Adam-like optimizer. $M_t$ represents the first moment, and $\alpha_t = (\sqrt{V_t} + \epsilon)^{-1}$ is the per-parameter adaptive learning rate derived from the second moment $V_t$. The key lies in the scale factor $S_t$, which is computed by partitioning the model's parameters according to its architecture and then deriving statistics from these structurally meaningful groups. The core idea is to factorize the learning rate for each parameter into components that capture adaptation at different architectural granularities. In particular, by allowing $\alpha_t$ to vary within blocks while normalizing it via a partition-aware $S_t$ that is uniform across blocks, SAC simultaneously (i) realizes heterogeneous, adaptive rates inside blocks and (ii) enforces uniform scaling constraints across blocks. Concretely, SAC couples multi-resolution LR modulation aligned with model topology (parameter $\rightarrow$ group $\rightarrow$ layer $\rightarrow$ block) with a dual-objective design that preserves cross-layer coherence without sacrificing locality. To precisely define these structural groups and compute $S_t$, we first introduce our method for *Architecture-Aware Parameter Partitioning*.

## 2.1 ARCHITECTURE-AWARE PARAMETER PARTITIONING

As aforementioned, SAC requires a structured view of the model. We treat neural network topology as a multi-resolution index over its parameter space $\Theta$. This is achieved by partitioning the model parameters along two primary axes: network depth (layers) and intra-layer functional roles (blocks).

Let $L$ be the ordered set of layers within the model (*e.g.*, Transformer blocks, embedding layers, and output head). Each layer $l \in L$ is composed of macro blocks $B_l$, such as a token mixer (self-attention) and a channel mixer (MLP). For each block $b \in B_l$, we define a parameter set $P_{l,b} \subset \Theta$ containing its core weight matrices (*e.g.*, $W_q, W_k, W_v, W_o, W_{\text{in}}, W_{\text{out}}$), excluding scalar parameters like biases. This yields a complete and disjoint partition of all model weights as:

$$\Theta = \biguplus_{l \in L} \biguplus_{b \in B_l} P_{l,b}, \tag{2}$$

This provides well-defined scopes for computing statistics, from fine-grained (parameter-wise) to coarse-grained (block-wise, layer-wise, group-/subspace-wise (*e.g.*, heads, rows/columns, low-rank

subspaces of $W \in \mathbb{R}^{m \times n}$)). It also admits constant-time indexers $\pi_{\text{layer}}, \pi_{\text{block}}$ which aligns with distributed training schemes (*e.g.*, tensor parallelism) and allows for efficient aggregation of statistics with negligible overhead, adding only $O(|L| + \sum_l |B_l|)$ scalars of state.

Subsequently, we attach *Structured Learning Rates (SLR)* to this topology. The effective learning rate $\alpha_\theta$ for each parameter $\theta \in P_{l,b}$ is factorized as a product of hierarchical components as:

$$\alpha_\theta = \eta \cdot c_l \cdot s_b \cdot r_\theta, \tag{3}$$

where $r_\theta$ captures *within-block* heterogeneity from local curvature/scale estimates (e.g., per-head or low-rank row/column groups); $s_b$ imposes block-level calibration distinguishing token vs. channel mixers; and $c_l$ enforces *cross-layer* coherence via normalization constraints (e.g., $\mathbb{E}_{\theta \in P_{l,b}}[r_\theta] = 1$). This factorization separates global consistency, regulated by $(s_b, c_l)$, from fine-grained adaptation, tracked by $r_\theta$, while regulating signal magnitudes for stability and hardware efficiency.

## 2.2 MULTI-RESOLUTION SCALE FACTORS

With the model partitioned, SAC computes scale factors to achieve two complementary objectives: (i) *Inter-layer Uniformity*: Maintain comparable signal strength and update magnitudes across all layers to promote stable training in deep networks; and (ii) *Intra-layer Heterogeneity*: Allow different functional blocks within a layer (*e.g.*, attention vs. MLP) to adapt at different rates according to their specific roles and gradient statistics. To achieve this, we introduce two factors, $c_l$ and $s_{l,b}$, derived from gradient statistics at step $t$. Let $g_{l,b}^t \in \mathbb{R}^{d_{l,b}}$ be the gradient vector for the block $b$ in layer $l$; write $g_l^t$ be the concatenated gradient for the entire layer $l$, and $g^t$ be the gradient for the entire model. We use the median absolute deviation (MAD) as a robust, outlier-resistant measure of statistical dispersion:

$$\mathcal{D}(X) = \text{median}\Big( |X - \text{median}(X)| \Big), \tag{4}$$

---

**Algorithm 1** SAC Optimizer Wrapper

**Require:** Global learning rate $\eta$, decay rates $\beta_1, \beta_2, \varepsilon$, weight decay $\lambda$, scale bounds $[S_{\min}, S_{\max}]$.
**Require:** Parameters $\Theta = \{\theta_i\}$, gradients $G = \{G_i\}$.
**Ensure:** Updated model parameters $\Theta$.
1: Initialize: time step $t \leftarrow 0$, $m, v \leftarrow 0$, $\mathcal{M}, \mathcal{S} \leftarrow \emptyset$
    **Recursive Scale Factor Computation**
2: $t \leftarrow t + 1$
3: $\delta_{\text{global}} \leftarrow \text{GlobalMAD}(\theta, G, \mathcal{S})$
4: **function** PROCESSPARAM$(\theta_i, G_i, l_i, b_i)$
5:     **If** first call: $\mathcal{M}[\theta_i] \leftarrow (l_i, b_i, \dim(\theta_i) > 1)$   ▷ Cache parameter metadata
6:     **If** first call: $\mathcal{S}[l_i][b_i] \leftarrow \mathcal{S}[l_i][b_i] \cup \{\theta_i\}$  ▷ Group by layer/block/param
7:     $\delta_l \leftarrow \text{LayerMAD}(l_i, \mathcal{S})$
8:     $\text{mad}_{l,b} \leftarrow \text{BlockMAD}(l_i, b_i, \mathcal{S})$
9:     $s_{l,b} \leftarrow \text{mean}\big(\log(1 + |G - \mu|/\text{mad}_{l,b})\big)$
10:     $s_i \leftarrow \max\big(S_{\min}, \min(S_{\max}, (\delta_{\text{global}}/\delta_l) \cdot s_{l,b})\big)$ ▷ Hierarchical scaling
11:     $s_i \leftarrow \begin{cases} s_i & \text{if } \dim(\theta_i) > 1 \\ 1 & \text{otherwise} \end{cases}$
    *(e.g., AdamW)*
12:     $m_i \leftarrow \beta_1 m_i + (1 - \beta_1)G_i$
13:     $v_i \leftarrow \beta_2 v_i + (1 - \beta_2)G_i^2$
14:     $\hat{m}_i \leftarrow m_i/(1 - \beta_1^t)$
15:     $\hat{v}_i \leftarrow v_i/(1 - \beta_2^t)$
16:     $\theta_i \leftarrow \theta_i - \eta \cdot s_i \cdot \frac{\hat{m}_i}{\sqrt{\hat{v}_i} + \varepsilon} - \eta \cdot \lambda \cdot \theta_i$
17:     **return** $\Theta \leftarrow \{\theta_i\}$
18: **end function**

---

**Layer Factor for Uniformity.** To equalize update scales across depth, we compute a layer-specific factor $c_l^t$ that compares the dispersion of the layer's gradient $g_l^t$ to that of the global gradient $g^t$ as:

$$c_l^t = \Big(\frac{\mathcal{D}(g^t) + \varepsilon}{\mathcal{D}(g_l^t) + \varepsilon}\Big)^\gamma, \qquad \gamma \in [0, 1], \tag{5}$$

where a small constant $\varepsilon > 0$ is used for numerical stability. This factor is then optionally smoothed over time using an exponential moving average (EMA) with decay rate $\rho$ to produce $\tilde{c}_l^t$ as:

$$\tilde{c}_l^t = (1 - \rho)\,\tilde{c}_l^{t-1} + \rho\,c_l^t, \qquad \rho \in (0, 1]. \tag{6}$$

Note that layers with smaller relative dispersion ($\mathcal{D}(g_l^t) \ll \mathcal{D}(g^t)$) receive a factor $\tilde{c}_l^t > 1$, effectively mitigating depth imbalance with negligible computational and communication overhead.

**Block Factor for Heterogeneity.** Within each layer $l$, we aim to allocate the "update budget" based on the relative importance or signal strength of each block. Thus, we first calculate a scalar statistic $\phi_{l,b}^t$ for each block, such as the logarithm of its gradient Root Mean Square (RMS), as:

$$\phi_{l,b}^t = \log(\text{RMS}(g_{l,b}^t) + \varepsilon), \qquad \text{RMS}(x) = \sqrt{\frac{1}{m}\sum_{i=1}^m x_i^2}, \;\; m = \dim(x), \tag{7}$$

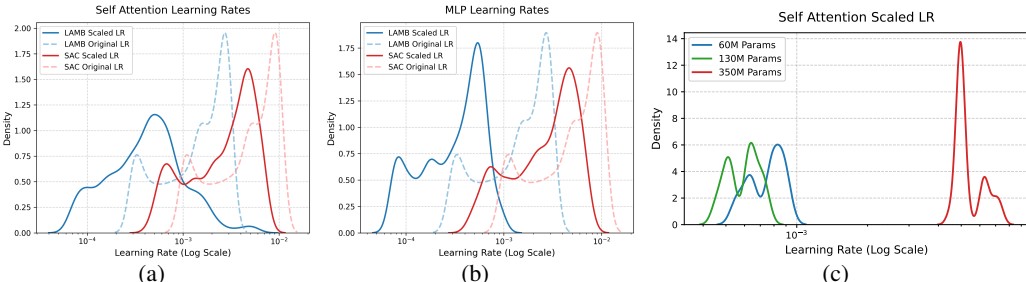

Figure 3: **Analysis of SAC-Modulated Learning Rate Distributions** with LLaMA-350M pre-trained on C4. **(a–b)** SAC applies distinct modulation to different blocks. For a 130M LLaMA, kernel density estimates of per-parameter learning rates in *Self-Attention* (a) and *MLP* (b). Solid curves show effective (scaled) rates under SAC and LAMB; dashed curves show the corresponding original step sizes from the AdamW inner optimizer. **(c)** For Self-Attention blocks, SAC-scaled learning rate distributions at step 10,000 show a clear size-dependent shift as model size increases, where the rates for 60M/130M concentrate below $10^{-3}$, while 350M's distribution is displaced toward higher rates.

We then employ a budgeted softmax with temperature $\beta \geq 0$ to assign multiplicative weights and derive the block-specific scale factors:

$$s_{l,b}^t \;=\; |B_l|\;\frac{\exp\!\big(\beta\,\phi_{l,b}^t\big)}{\sum_{b' \in B_l} \exp\!\big(\beta\,\phi_{l,b'}^t\big)}. \tag{8}$$

This ensures that the factors average to one within each layer as $\frac{1}{|B_l|}\sum_{b \in B_l} s_{l,b}^t = 1$. As such, blocks with stronger signals receive $s_{l,b}^t > 1$ while others receive $s_{l,b}^t < 1$, redistributing learning capacity without altering the layer's average update magnitude, which is illustrated in Figure 3.

**Composition & Scaling.**    The layer and block factors, $\tilde{c}_l^t$ and $s_{l,b}^t$ are composed multiplicatively to form the complete architectural scale factor tensor $S_t$ from the main update rule (Eq. 1). In practice, this is implemented as a straightforward plug-and-play multiplicative correction. The effective learning rate $\eta_{l,b}^t$ for all parameters within block $b$ of layer $l$ is modulated as:

$$\eta_{l,b}^t \;\leftarrow\; \eta_{l,b}^t\,\big(\tilde{c}_l^t\,s_{l,b}^t\big), \tag{9}$$

This is compatible with standard techniques like mixed-precision and distributed training. Furthermore, it preserves important sanity cases: if all layers have similar gradient dispersion, $c_l^t \approx 1$, and if all blocks within a layer are statistically similar, block factors $s_{l,b}^t$ also approach 1, thereby recovering the behavior of the base optimizer. Hyperparameters $(\gamma, \beta, \rho)$ serve as intuitive knobs to control inter-layer uniformity and intra-layer contrast, offering a path toward stable training.

## 3 EXPERIMENTS

### 3.1 EXPERIMENTAL SETUP

To rigorously evaluate the effectiveness and versatility of SAC, we conducted experiments on various public datasets, including LLM, Visual Question Answering (VQA), and multimodal LLM (MLLM) benchmarks. **(1) LLM Pre-training** with short/long sequence modeling: We used the `en` subset of the C4 dataset (a large cleaned web corpus from Common Crawl filtered for safety (Köpf et al., 2023)) for LLaMA pre-training, while adopting the 100B subset of the FineWeb-Edu dataset (Penedo et al., 2024) for long-sequence modeling with Gated DeltaNet variants (Yang et al., 2025). **(2) Image Classification** with various architectures: Sharing the Metaformer macro designs (Yu et al., 2024), we trained the typical Transformer, CNNs, and hybrid models from scratch on CIFAR-100 (Krizhevsky et al., 2009) and ImageNet-1K (Krizhevsky et al., 2012) datasets to provide a standard measurement of generalization to different networks. **(3) PEFT on Commonsense Reasoning:** Leveraging the LLM-Adapters framework (Hu et al., 2023), we evaluated SAC's compatibility and performance with pre-trained models and PEFT methods across 8 Commonsense Reasoning (CS) datasets: BoolQ (Clark et al., 2019), PIQA (Bisk et al., 2020), SIQA (Sap et al., 2019),

Table 1: **LLaMA Pre-training Comparison on the C4 Dataset** with model sizes ranging from 60M to 1B. We report three key metrics: the validation perplexity (PPL)↓, GPU memory (Mem.)↓ (including model weights and optimization states), and the averaged running Time (s)↓ of optimizer step. For all metrics, lower is better. The practical hyperparameters are clearly tuned and reported for all optimizers. **Bold** denotes the best results in each category, while green and red types denote the performance gains↓ of SAC (blue background) over related baselines (gray background). Note that **SAC+AdamW** achieves the best performance over all compared optimizers.

| Optimizer | Venue | Betas | Eps. | 60M | | | 130M | | | 350M | | | 1B | | |
|---|---|---|---|---|---|---|---|---|---|---|---|---|---|---|---|
| | | | | PPL | #M(G) | Time(s) | PPL | #M(G) | Time(s) | PPL | #M(G) | Time(s) | PPL | #M(G) | Time(s) |
| AdamW | ICLR'19 | (0.9, 0.99) | 1e-8 | 29.19 | 0.25 | 0.0018 | 22.64 | 0.55 | 0.0023 | 16.97 | 1.43 | 0.0045 | 14.40 | 5.11 | 0.0762 |
| Adabelief | NeurIPS'19 | (0.9, 0.999) | 1e-12 | 29.49 | 0.46 | 0.0099 | 22.92 | 1.04 | 0.0156 | 17.46 | 2.80 | 0.0614 | 16.85 | 10.1 | 0.2448 |
| Adamp | ICLR'21 | (0.9, 0.98) | 1e-8 | 29.34 | 0.25 | 0.0263 | 22.52 | 0.55 | 0.0397 | 17.04 | 1.43 | 0.1139 | 14.41 | 5.11 | 0.2836 |
| LAMB | ICLR'20 | (0.9, 0.99) | 1e-6 | 29.08 | 0.25 | 0.0168 | 22.57 | 0.55 | 0.0274 | 16.89 | 1.43 | 0.0897 | 15.32 | 5.11 | 0.2269 |
| Nadam | ICLR'18 | (0.9, 0.99) | 1e-8 | 32.75 | 0.25 | 0.0029 | 24.04 | 0.55 | 0.0040 | 17.57 | 1.43 | 0.0065 | 16.48 | 5.11 | 0.0879 |
| Radam | ICLR'20 | (0.9, 0.99) | 1e-8 | 29.23 | 0.25 | 0.0024 | 22.67 | 0.55 | 0.0031 | 16.94 | 1.43 | 0.0053 | 14.30 | 5.11 | 0.0994 |
| Adan | TPAMI'23 | (0.9, 0.92, 0.99) | 1e-8 | 29.40 | 0.46 | 0.0042 | 22.30 | 1.04 | 0.0041 | 17.01 | 2.80 | 0.0158 | 14.70 | 10.1 | 0.1787 |
| Prodigy | ICML'23 | (0.9, 0.95) | 1e-8 | 32.33 | 0.46 | 0.0141 | 29.56 | 1.04 | 0.0257 | 17.96 | 2.80 | 0.0814 | 14.94 | 10.1 | 0.2298 |
| MARS+AdamW | ICML'25 | (0.9, 0.99) | 1e-8 | 29.10 | 0.32 | 0.0147 | 22.26 | 0.75 | 0.0290 | 16.65 | 2.06 | 0.0804 | 14.76 | 7.48 | 0.2333 |
| SGG+AdamW | ACL'25 | (0.9, 0.99) | 1e-8 | 29.98 | 0.46 | 0.0392 | 22.13 | 1.04 | 0.0631 | 16.97 | 1.43 | 0.0714 | 14.34 | 4.77 | 0.3526 |
| SAC+AdamW | Ours | (0.9, 0.99) | 1e-8 | **28.63** | 0.25 | 0.0169 | **21.85** | 0.55 | 0.0213 | **16.16** | 1.43 | 0.0401 | **13.58** | 5.11 | 0.1089 |
| △Gains | | | | -0.56 | +0 | +0.0152 | -0.79 | +0 | 0.0190 | -0.81 | +0 | 0.0363 | -0.82 | +0 | 0.0329 |
| Adam8bit | ICLR'22 | (0.9, 0.99) | 1e-8 | 29.47 | 0.14 | 0.0091 | 22.74 | 0.30 | 0.0189 | 17.35 | 0.76 | 0.0652 | 14.49 | 2.66 | 0.2286 |
| Adam-mini | ICLR'25 | (0.9, 0.99) | 1e-8 | 29.63 | 0.14 | 0.0106 | 23.08 | 0.30 | 0.0152 | 19.25 | 0.75 | 0.0599 | 16.44 | 2.62 | 0.1868 |
| Adafactor | ICML'18 | (0.9,) | 1e-30 | **29.07** | 0.24 | 0.0059 | 22.38 | 0.61 | 0.0082 | 16.96 | 1.53 | 0.0447 | 16.25 | 6.65 | 0.1725 |
| CAME | ACL'23 | (0.9, 0.98) | 1e-8 | 29.26 | 0.18 | 0.0068 | 22.55 | 0.38 | 0.0084 | 16.84 | 1.08 | 0.0451 | 15.76 | 3.83 | 0.1794 |
| APOLLO | MLSys'25 | (0.9, 0.99) | 1e-6 | 29.82 | 0.24 | 0.0061 | **22.18** | 0.52 | 0.0090 | **16.54** | 1.22 | 0.0453 | **13.91** | 4.38 | 0.1809 |
| Lion | arXiv'23 | (0.9, 0.98) | − | 34.80 | 0.14 | 0.0049 | 24.95 | 0.30 | 0.0057 | 18.84 | 0.75 | 0.0400 | 17.01 | 2.62 | 0.1684 |
| Sophia | arXiv'23 | (0.9, 0.99) | 1e-8 | 35.14 | 0.25 | 0.0080 | 25.09 | 0.55 | 0.0105 | 18.42 | 1.43 | 0.0478 | 17.62 | 5.11 | 0.1843 |
| MARS+Lion | ICML'25 | (0.9, 0.98) | 1e-8 | 31.50 | 0.32 | 0.0139 | 25.02 | 0.75 | 0.0247 | 18.36 | 2.06 | 0.0753 | 16.94 | 7.48 | 0.1804 |
| SAC+Adam-mini | Ours | (0.9, 0.99) | 1e-8 | 29.49 | 0.14 | 0.0131 | 22.62 | 0.30 | 0.0157 | 16.66 | 0.75 | 0.0605 | 14.23 | 2.62 | 0.1873 |
| △Gains | | | | -0.14 | +0 | 0.0025 | -0.46 | +0 | 0.0005 | -2.59 | +0 | 0.0006 | -2.21 | +0 | 0.0005 |
| Shampoo | arXiv'18 | (0.9, 0.999) | 1e-8 | 29.30 | 0.18 | 0.0364 | 22.01 | 0.35 | 0.0526 | 16.71 | 1.37 | 0.1465 | 14.34 | 4.77 | 0.8762 |
| Muon (kimi) | arXiv'25 | (0.9, 0.95) | 1e-8 | 28.91 | 0.14 | 0.0336 | 22.19 | 0.30 | 0.0486 | 16.72 | 0.75 | 0.1370 | 14.52 | 2.62 | 0.8870 |
| SOAP | arXiv'24 | (0.9, 0.95) | 1e-8 | **28.60** | 0.17 | 0.0747 | 22.15 | 0.34 | 0.1028 | 16.79 | 1.35 | 0.1943 | 14.58 | 4.72 | 0.9205 |
| MARS+Shampoo | ICML'25 | (0.9, 0.99) | 1e-8 | 29.13 | 0.32 | 0.0491 | **21.96** | 0.75 | 0.0768 | **16.49** | 2.06 | 0.1537 | **13.75** | 7.48 | 0.8823 |
| SAC+Shampoo | Ours | (0.9, 0.999) | 1e-8 | 29.22 | 0.18 | 0.0376 | **21.96** | 0.35 | 0.0541 | 16.61 | 1.37 | 0.1481 | 14.07 | 4.77 | 0.8785 |
| △Gains | | | | -0.08 | +0 | 0.0012 | -0.05 | +0 | 0.0015 | -0.09 | +0 | 0.0016 | -0.27 | +0 | 0.0023 |

HellaSwag (Zellers et al., 2019), WinoGrande (Sakaguchi et al., 2021), ARC (ARC-Easy and ARC-Challenge) (Clark et al., 2018), and OBQA (Mihaylov et al., 2018). **(4) MLLM Validation:** (i) VQA benchmarks such as GQA (Hudson & Manning, 2019), TextVQA (Singh et al., 2019), SciVQA$^I$ (evaluation on the imageset of ScienceVQA) (Lu et al., 2022), VQAv2 (Goyal et al., 2017), and Vizwiz (Gurari et al., 2018). (ii) MLLM evaluation benchmarks including POPE (F1 score) (Li et al., 2023b), MMBench (Liu et al., 2025), MMBench-Chinese (MMBench$^{CN}$) (Liu et al., 2025), SEED$^I$ (Li et al., 2023a), and MME (Perception) (Yin et al., 2023). As for implementations, we applied SAC as an optimizer wrapper upon popular optimizer baselines (AdamW, Adam-mini, and Shampoo) in PyTorch, ensuring compatibility with existing optimizers through minimal code integration. Its key hyperparameters were empirically tuned, allowing for robust use of the default values for optimal accuracy-efficiency trade-off.

## 3.2 COMPARISON RESULTS WITH TRAINING FROM SCRATCH

We validate our SAC in various LLM tasks, including pre-training, supervised fine-tuning (SFT), and parameter-efficient fine-tuning (PEFT). SAC consistently improves performance with negligible extra cost, showcasing its potential as a versatile optimizer wrapper for effective LLM training.

**Pre-training on C4 Benchmark.** Following Galore (Zhao et al., 2024a), we reproduce LLaMA pre-training in Table 1 with a comprehensive comparison of popular LLM optimizers. There are lots of optimizers that achieve a well-done performance on the C4 dataset, since they heavily rely on the optimal hyperparameters. To ensure a fair evaluation, we further conducted experiments under relatively standardized and controlled settings across different model scales. The remaining open question is the trade-off between performance (perplexity) and parallelism. Memory-efficient optimizers, *e.g.*, MARS and Muon, achieve superior performance by employing complex matrix operations, but this inevitably leads to reduced parallelism. In contrast, others like AdamW offer higher throughput but may yield slightly lower performance. The AdamW runs faster around ×32 than Muon on LLaMA 350M. As a lightweight wrapper, SAC is designed to navigate this trade-off,

Table 2: **Image classification** on CIFAR-100 and ImageNet-1K. Comparing SAC+AdamW with adaptive LR optimizers upon various architectures that are trained from scratch and evaluated by top-1 accuracy (%)↑, **bold**, (underline) denote the best and second results, while green types denote the performance gains↑ upon AdamW baseline.

| Optimizer | CIFAR-100 | | | | ImageNet-1K | |
|---|---|---|---|---|---|---|
| | DeiT-S | Swin-T | CNX-T | CA-S12 | R-50 | DeiT-S |
| AdamW | 72.15 | 81.30 | 83.52 | 83.60 | 79.88 | 80.38 |
| NAdam | 72.75 | 81.80 | 83.06 | 82.83 | 78.16 | 78.26 |
| AdamP | 71.55 | 80.91 | 84.47 | 83.40 | 79.83 | 79.71 |
| Adan | **76.33** | 83.35 | **84.65** | **84.89** | 79.79 | 80.81 |
| AdaFactor | 74.02 | 80.36 | 82.82 | 82.36 | 79.71 | 79.98 |
| AdaBelief | 70.66 | 80.98 | 83.31 | 83.56 | 79.48 | 75.32 |
| RAdam | 72.41 | 79.84 | 82.18 | 82.35 | 77.96 | 78.54 |
| LAMB | 75.39 | **83.47** | 84.13 | 83.74 | 79.84 | 80.23 |
| LION | 74.57 | 81.84 | 82.29 | 79.59 | 77.36 | 78.78 |
| Sophia | 71.47 | 80.61 | 83.76 | 82.96 | 79.32 | 79.65 |
| SAC+AdamW | 76.05 | 83.43 | 84.58 | 84.58 | **80.12** | **80.87** |
| △ Gains | +3.90 | +2.13 | +1.06 | +0.98 | +0.24 | +0.49 |

Table 3: **Long-sequence modeling** with Gated DeltaNet variants of 1.3B on FineWeb-Edu dataset (100B). Perplexity (PPL↓) on Wikitext and LAMBDADA datasets, and zero-shot top-1 accuracy (%)↑ on commonsense reasoning datasets are reported, where SAC+AdamW shows consistent performance improvements.

| Task | Metric | Gated DeltaNet | | | Gated DeltaNet-H1 | | |
|---|---|---|---|---|---|---|---|
| | | AdamW | Muon | SAC | AdamW | Muon | SAC |
| Wiki. | PPL↓ | 16.42 | 16.31 | **15.95** | 16.07 | 15.98 | **15.70** |
| LMB. | | 12.17 | 12.12 | **12.03** | 12.12 | 12.08 | **11.98** |
| LMB. | | 46.65 | 46.75 | **46.87** | 47.73 | 47.21 | **48.03** |
| PIQA | | 72.25 | 72.38 | **72.45** | 72.57 | 72.60 | **72.65** |
| Hella. | | 55.76 | 55.82 | **56.01** | 56.53 | 56.64 | **56.78** |
| Wino. | Acc.↑ | 57.45 | 57.61 | **57.80** | 58.40 | 58.39 | **58.51** |
| ARC-E | | 71.21 | 71.25 | **71.36** | 71.75 | 71.92 | **72.13** |
| ARC-C | | 38.39 | 38.44 | **38.52** | 40.10 | 40.25 | **40.28** |
| SIQA | | 40.63 | 40.70 | **40.85** | 41.40 | 41.35 | **41.62** |
| BoolQ | | 60.24 | 60.36 | **61.29** | 63.21 | 63.28 | **63.31** |
| AVG. | Acc.↑ | 55.32 | 55.41 | **55.64** | 56.46 | 56.46 | **56.66** |

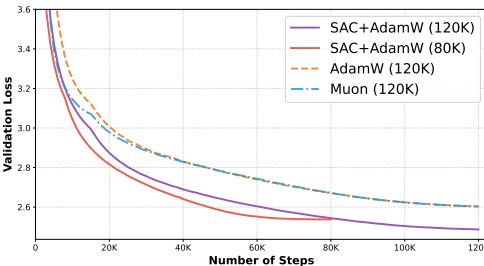

Figure 4: Validation loss curve of LLaMA 3B pre-trained on C4, where both SAC+AdamW with long (120K) and short (80K) training budgets show better performances and faster convergences than AdamW and the recent Muon.

Table 4: **Pre-training LLaMA 3B** on C4 dataset for 150K iterations with validation perplexity (PPL), optimization memory (weights and optimization states), and average running times of the optimization step are reported. **Bold**, green, and red types denote the best results, performance gains, and throughput decreases compared to the baselines.

| Optimizer | Memory | Time | 40K | 80K | 120K | 160K |
|---|---|---|---|---|---|---|
| AdamW | 10.2G | 0.203s | 16.97 | 14.45 | 13.51 | 13.45 |
| Adam-mini | 5.2G | 0.249s | 20.61 | 15.07 | 14.43 | 14.26 |
| Muon | 5.2G | 1.023s | 16.91 | 14.46 | 13.5 | 13.44 |
| APOLLO | 5.2G | 0.240s | 17.62 | 14.59 | 14.36 | 13.81 |
| SAC+Adam-mini | 5.2G | 0.251s | 16.58 | 13.67 | 13.15 | 12.92 |
| △Gains | +0G | +0.002 | -4.03 | -1.40 | -1.28 | -1.34 |
| SAC+AdamW | 10.2G | 0.235s | **14.73** | **12.73** | **12.02** | **12.00** |
| △Gains | +0G | +0.032 | -2.24 | -1.72 | -1.49 | -1.45 |

enhancing base optimizers to improve performance with a lower time cost than MARS (lower 0.69 Mem (G) and 0.0227 time (s) on LLaMA 130M). Moreover, SAC can be aggregated with different optimizers to achieve a trade-off between performance and cost for specific tasks. *e.g.*, SAC+AdamW achieves the **16.16** PPL with no extra memory, which is more efficient than MARS+AdamW.

**Long Sequence Modeling Pre-training on C4 Benchmark.** For fair comparison, we follow Gated DeltaNet (GDN) (Yang et al., 2025) to train the pure GRN and the hybrid variant (H1) of GRN and Sliding-window Attention (H1) modules under identical conditions with 1.3B parameters on 100B tokens sampled from the FineWeb-Edu dataset. We follow the default optimization setups, *i.e.*, AdamW optimizer with a peak learning rate of 4e-4, a weight decay of 0.1, and the epsilon of 1e-15. The learning rate follows a cosine annealing schedule with a 1B token warm-up period and a batch size of 0.5M tokens. As shown in Table 3, SAC consistently improves the pre-training PPL and zero-shot accuracy of AdamW and Muon on CS reasoning datasets. View Appendix A for details.

**Image Classification with Vision Architectures.** Following BOCB benchmarks (Li et al., 2024b), we verify the generalization abilities of the SAC wrapper with typical vision backbones on CIFAR-100 and ImageNet-1K with 200 and 300 epochs of training with advanced setups (Wightman et al., 2021). The benchmarked architectures include CNNs (ResNet-50 (He et al., 2016) and ConvNeXt-T (Liu et al., 2022)), Transformers (DeiT-S (Touvron et al., 2021) and Swin-T (Liu et al., 2021)), hybrid model (CAFormer-S12 (Yu et al., 2024)), which are shown in Table 2 with abbreviations of model names. It provides strong evidence that SAC+AdamW could be migrated to heterogeneous networks by applying the architectural constraints at the macro-design levels.

**PEFT on Commonsense Reasoning.** Following LLM-Adapters, we assess SAC in CS tasks with top-1 accuracy and GPU memory, where LLaMA-7B is fine-tuned by AdamW+LoRA ($r = 32$) on a

Table 5: **LLaMA-7B PEFT** on commonsense reasoning datasets with top-1 accuracy (%) ↑, where LoRA+SAC is compared against PEFT baselines and memory-efficient optimizers, **bold** and green types denote the best results and performance gains↑ compared to the LoRA baseline.

| Optimizer | PEFT | Params. | Memory | BoolQ | PIQA | SIQA | HellaS. | WinoG. | ARC-E | ARC-C | OBQA | Average |
|---|---|---|---|---|---|---|---|---|---|---|---|---|
| AdamW | Prefix | low-rank | 0.05G | 64.3 | 76.8 | 73.9 | 42.1 | 72.1 | 72.9 | 54.0 | 60.6 | 64.6 |
| AdamW | Series | low-rank | 0.42G | 63.0 | 79.2 | 76.3 | 67.9 | 75.7 | 74.5 | 57.1 | 72.4 | 70.8 |
| AdamW | Parallel | low-rank | 1.49G | 67.9 | 76.4 | **78.8** | 69.8 | 78.9 | 73.7 | 57.3 | 75.2 | 72.3 |
| AdamW | LoRA | low-rank | 0.35G | 68.9 | 80.7 | 77.4 | 78.1 | 78.8 | 77.8 | 61.3 | 74.8 | 74.7 |
| AdamW | DoRA | low-rank | 0.26G | 69.7 | 83.4 | 78.6 | **87.2** | 81.0 | 81.9 | **66.2** | 79.2 | **78.4** |
| AdamW | Fira | low-rank | 0.26G | 69.4 | 82.6 | 78.0 | 76.8 | **81.2** | **82.2** | 64.4 | **80.8** | 76.9 |
| GaLore | – | full-rank | 0.26G | 69.5 | 82.0 | 75.1 | 32.2 | 18.0 | 80.7 | 65.8 | 78.0 | 62.7 |
| APOLLO (SVD) | – | full-rank | 0.37G | 69.4 | 82.2 | 78.7 | 68.6 | 80.6 | 81.8 | **66.2** | 79.9 | 75.9 |
| SGG+AdamW | LoRA | low-rank | 0.35G | 70.3 | 83.6 | **78.8** | 81.7 | 80.9 | 81.5 | 65.3 | 79.0 | 77.6 |
| SAC+AdamW | LoRA | low-rank | 0.35G | **70.5** | **83.7** | 78.5 | 81.6 | 81.1 | 81.7 | 65.4 | 79.3 | 77.7 |
| △Gains | | | +0G | 1.6 | 3.0 | 1.1 | 3.5 | 2.3 | 3.9 | 4.1 | 4.5 | 3.0 |

Table 6: **Full Comparison Results on LLaVA-v1.5 7B Benchmark**. Compared with their counterparts, top-1 accuracy (%) ↑ is reported. AVG is the average result of the nine benchmarks, except for MME. Green types denote the performance gains↑ of SAC over baselines.

| Optimizer | Image Question Answering | | | | | Benchmarks | | | | | AVG. | Gain |
|---|---|---|---|---|---|---|---|---|---|---|---|---|
| | VQAv2 | GQA | VizWiz | SciQA$^I$ | TextVQA | MME | MMBench | MMBench$^{CN}$ | POPE | SEED$^I$ | | |
| AdamW | 78.5 | 62.0 | 50.0 | 66.8 | 58.2 | 1510.7 | 64.3 | 58.3 | 85.8 | 66.2 | 65.56 | – |
| Adafactor | 79.29 | 62.7 | 48.15 | 69.76 | 57.1 | 1462.5 | 66.15 | 60.39 | 86.11 | 66.79 | 66.27 | +0.71 |
| LAMB | 71.78 | 51.0 | 45.54 | 66.19 | 50.81 | 1309.99 | 54.03 | 49.48 | 82.76 | 55.64 | 58.58 | -6.98 |
| RAdam | 79.15 | 62.49 | 51.92 | 69.46 | 57.77 | 1475.23 | 66.4 | 61.25 | 86.24 | 67.27 | 66.88 | +1.32 |
| NAdam | 79.2 | 62.53 | 48.77 | 69.16 | 57.6 | 1467.68 | 66.49 | 60.99 | 86.10 | 66.59 | 66.38 | +0.82 |
| Adan | 78.77 | 62.17 | 48.39 | 70.3 | 57.74 | 1491.08 | 66.06 | 60.22 | 86.08 | 66.39 | 66.23 | +0.67 |
| Shampoo (Muon) | 79.34 | 62.67 | 50.34 | 69.06 | 57.71 | 1461.7 | 67.1 | 59.87 | 85.94 | 67.01 | 66.56 | +1.0 |
| SOAP | 79.36 | 62.51 | 47.85 | 69.71 | 57.98 | 1475.09 | 66.58 | 60.13 | 86.24 | 67.43 | 66.42 | +0.86 |
| Sophia | 78.29 | 61.48 | 49.8 | 69.56 | 56.44 | 1476.13 | 66.75 | 60.13 | 85.87 | 65.49 | 65.98 | +0.42 |
| LION | 78.98 | 62.28 | 48.81 | 69.91 | 56.7 | 1517.03 | 66.66 | 60.39 | 86.52 | 66.46 | 66.30 | +0.74 |
| CAME | 78.62 | 62.24 | 45.32 | 67.58 | 52.86 | 1419.53 | 64.69 | 52.14 | 86.33 | 65.99 | 63.97 | -1.59 |
| SGD | 74.55 | 56.29 | 40.65 | 68.27 | 53.72 | 1358.18 | 60.22 | 53.09 | 84.11 | 60.9 | 61.31 | -4.25 |
| MARS+AdamW | 79.25 | 62.82 | 49.24 | 69.11 | 56.43 | 1451.05 | 66.75 | 59.45 | 86.14 | 67.46 | 66.29 | +0.73 |
| MARS+Shampoo | 78.43 | 62.48 | 48.63 | 69.01 | 55.91 | 1426.37 | 66.88 | 59.86 | 85.9 | 67.45 | 66.06 | +0.5 |
| SGG+AdamW | 79.31 | 62.65 | 49.91 | 69.76 | 57.61 | 1462.69 | 66.32 | 60.48 | 85.92 | 67.18 | 66.57 | +1.01 |
| SGG+Adafactor | 79.2 | 62.78 | 50.63 | 69.71 | 57.32 | 1445.5 | 66.24 | 60.74 | 85.91 | 66.26 | 66.53 | +0.97 |
| SGG+Shampoo | 79.36 | 62.79 | 50.56 | 69.26 | 57.67 | 1451.57 | 66.06 | 59.27 | 86.27 | 67.38 | 66.51 | +0.95 |
| SAC+AdamW | 79.34 | 62.69 | 51.38 | 69.11 | 57.44 | 1480.05 | 66.4 | 60.99 | 86.53 | 67.19 | 66.79 | +1.23 |
| SAC+Shampoo | 79.35 | 62.7 | 50.74 | 69.41 | 58.0 | 1493.1 | 66.49 | 61.15 | 86.35 | 67.45 | 66.85 | +1.29 |

unified training dataset, followed by evaluation on each specific subset. As shown in Table 5, SAC improves LoRA by an average of **+2.9%**, with up to **+4.2%** gains on specific tasks like OBQA. It matches or surpasses PEFT baselines, e.g., Prefix (Li & Liang, 2021), Series(Houlsby et al., 2019), and Parallel (He et al., 2021), and more recent DoRA, GaLore, and Fira (Chen et al., 2024). Please view Table 5 and Appendix 5 for more details.

**Comparison Results with MLLMs**   Following the supervised fine-tuning setting of LLaVA-v1.5-7B. We use a pretrained Vicuna-v1.5-7B (Chiang et al., 2023) as decoder, a pretrained CLIP (Radford et al., 2021) as vision encoder, and a pretrained $2 \times$ MLP for aligning the visual to text. We choose some mainstream optimizers as the baseline. The results in Table 6 show that SAC boosts AdamW by **+1.23%** on average, and gains **+1.29%** on Muon optimizer. Please view Appendix A for details.

### 3.3 ABLATION STUDIES

Then, we further analyze the key designs of SAC with the experimental setup on C4, where P, L, and B denote param-wise, layer-wise, and block-wise learning rates, respectively.

**Granularity ablation**. The full hierarchy (*group + layer + block*) achieves the best PPL of 21.38 (130M) and 16.41 (350M) with a small but consistent edge over *layer+block* in Table 7, indicating a modest yet reliable gain from the

Table 7: Ablation of different granularity configurations.

| Configuration | 130M↓ | 350M↓ |
|---|---|---|
| P+B | 21.92 | 16.78 |
| P+L | 22.03 | 17.12 |
| P+L+B | 21.85 | 16.16 |
| G+L+B | 22.62 | 16.66 |
| P+G+L+B | **21.68** | **16.05** |

global controller. Removing block heterogeneity (layer-only) hurts (22.03/17.12); block-only helps (21.63/16.78) but still trails layer+block. Thus, block-wise allocation drives most gains; layer-wise equalization is synergistic; a lightweight group term stabilizes at scale.

**Dispersion statistic**. MAD (21.38/16.41) is the best choice verified in Table 8. As robustness to heavy tails decreases, PPL worsens: Huber/IQR (small penalties), mean-abs/L1 (larger), and std/L2–RMS/variance/max-abs (worst, up to +0.94/+0.80). This monotonic trend supports a rule of thumb: median-centered, robust dispersion preserves cross-layer normalization and within-layer budgets; outlier-amplifying measures mis-scale updates and harm coordination.

Table 8: Ablation of statistics for computing scale factors.

| Statistics for SAC | 130M↓ | 350M↓ |
|---|---|---|
| **MAD (median abs. dev.)** | **21.38** | **16.41** |
| Huber scale ($\delta = 1$) | +0.07 | +0.04 |
| IQR (Q3–Q1) | +0.11 | +0.06 |
| Mean absolute deviation | +0.16 | +0.08 |
| L1 norm (per-block) | +0.20 | +0.11 |
| Std. deviation | +0.29 | +0.22 |
| L2 norm (per-block RMS) | +0.41 | +0.37 |
| Variance | +0.77 | +0.64 |
| Max absolute (per-block) | +0.94 | +0.80 |

## 4 RELATED WORK

**Modern LLM Optimizers.** Modern LLM training confronts an "impossible quadrangle" balancing performance, memory, parallelism, theoretical compute. Existing optimizers trade these dimensions: curvature-aware methods (Muon (Jordan et al.), MARS (Yuan et al., 2024)) speed convergence but increase FLOPs and synchronization; memory-reduced approaches (Adafactor (Shazeer & Stern, 2018), GaLore (Zhao et al., 2024a), APOLLO (Zhu et al., 2024a)) save memory but risk accuracy or parallelism; quantized/small-state variants (BAdam (Luo et al., 2025), Adam-mini (Zhang et al., 2024), LISA (Pan et al., 2025)) improve throughput at potential accuracy cost. Scaling laws further reveal non-monotonic optimal hyperparameters, highlighting the brittleness of static settings (Zhao et al., 2024b; Zhang et al., 2025; Li et al., 2024a). Recent stabilizers (Huang et al., 2025; Luo et al., 2023) mitigate training drift via momentum resets or confidence cues. SAC addresses this by incorporating architecture-aware, adaptive constraints per-level, achieving a robust balance across the quadrangle while maintaining drop-in simplicity.

**Traditional Adam-Style Optimizers.** Classical adaptive methods like Adam follow a simple recipe: maintain exponential moving averages of past gradients to set per-parameter learning rates based on temporal constraints (Kingma & Ba, 2015). Variants differ mainly in the signal used to govern the step size, including gradient variance (Adam), gradient differences (*e.g.,* AdaBelief (Zhuang et al., 2020), Adan (Xie et al., 2023)), or sign-based schemes operate on quantized directions with momentum (*e.g.,* SignSGD (Bernstein et al., 2018), Lion (Chen et al., 2023)). These designs yield a low memory footprint, stable plug-and-play behavior, and broad applicability without intricate controller stacks (Li et al., 2024b). However, the core limitation lies in their reliance on the temporal dimension. By treating parameters as a flat, independent set, they ignore models' rich structure, *i.e.*, the hierarchy of layers, blocks, and functional components. We thus propose augmenting temporal constraint with an orthogonal set of architectural constraints, which retains the simplicity of Adam-style parameter update while improving efficiency and final accuracy at scale.

## 5 CONCLUSION

**Contribution.** This work presents Scaling with Architectural Constraints (SAC), a fresh optimization framework that leverages the inherent architectural hierarchy of deep neural networks to address the optimizer quadrilemma of performance, GPU memory, computational efficiency, and parallel scalability. By imposing constraints at block and layer levels, SAC achieves a globally coordinated training process. Extensive evaluations on language, vision, and multimodal benchmarks show that SAC consistently outperforms strong baselines, delivering improved convergence and performance. These show that architecture-aware optimization could be helpful for training complex deep models.

**Limitation and Future Work.** Despite its promising performance, SAC still has several limitations for future research. Key directions include moving beyond the current hand-designed constraints to methods that learn them dynamically, for example, via meta-learning. Moreover, a rigorous theoretical analysis would be beneficial to understand the mechanisms, particularly their effects on model conditioning. Finally, exploring the integration of SAC with more recent optimizers, such as Lion or Muon, could unlock further performance gains and push the frontier of large model training.

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

## DECLARATION OF LLM USAGE

We use the Large Language Models (LLMs) for this paper to serve one purpose: to aid and polish the paper writing. We use the LLMs in a very limited capacity, restricted to minor editing of grammar, phrasing, and readability. We do not involve the LLMs in designing the method, developing theoretical results, or conducting experiments.

## A    IMPLEMENTATION DETAILS

We provide details of task backgrounds, datasets, training & evaluation settings, and experiment results with more baselines for various LLM/MLLM downstream tasks (Wang, 2018; Hu et al., 2023; Liu et al., 2024b).

### A.1    LLM PRE-TRAINING ON C4

We conduct extensive pre-training on LLaMA-based language models using the C4 corpus—a rigorously cleaned derivative of Common Crawl that has become a standard benchmark for large-scale pre-training and word-representation learning. To mimic real-world training conditions, we adopt a no-repeat sampling protocol over a large data volume and scale model sizes up to 7B parameters. We also summarize the LLaMA architecture and the pre-training hyperparameters. Unless noted otherwise, hyperparameters are held fixed across sizes: maximum sequence length of 256 tokens and a token-batch of 131,072 tokens ($\approx$ 131k; 512 samples $\times$256 tokens). For all optimizers, we warm up the learning rate for the first $10\%$ of steps and then apply cosine annealing down to $10\%$ of the initial rate.

For learning-rate selection, we run a systematic sweep on models between 60M and 1B parameters over $\{1e-2, 2e-2, 1e-3, 3e-3\}$, choosing the best setting by validation perplexity. Notably, SAC exhibits strong hyperparameter robustness, maintaining competitive performance across sizes under a single learning rate. The complete results of the C4 pretraining are reported in Table 1. We include popular baselines from prior work—Adam, Adam-mini (Zhang et al., 2024), APOLLO (Zhu et al., 2024a), LoRA (Hu et al., 2021)—and we reproduce additional optimizers under the same experimental setup, including Adafactor (Shazeer & Stern, 2018), NAdam (Reddi et al., 2018), RAdam (Liu et al., 2020), Adan (Xie et al., 2023), LAMB (You et al., 2020), LION (Chen et al., 2023), CAME (Luo et al., 2023), and Muon (Jordan et al.). Looking ahead, the observed stability of SAC suggests lower tuning overhead as models scale beyond 7B and provides a practical recipe for web-scale pre-training under tight compute or data-throughput constraints.

Table A1: Details of the hyperparameters for the included optimizers and experiment settings.

| Method | AdamW | Shampoo |
|---|---|---|
| **Modules and datasets** | | |
| LLM | Vicuan-v1.5-7B | |
| Vision encoder | CLIP-L-336px | |
| Connector | 2$\times$MLP | |
| Pretrain data | `LCS-558K` | |
| SFT data | `llava-v1.5-mix665k` | |
| **Basic SFT settings** | | |
| Learning rate | $2e^{-5}$ | $2e^{-5}$ |
| Batch size | 64 | 64 |
| Betas | (0.9, 0.999) | (0.9, 0.999) |
| Epsilon | $1e^{-8}$ | $1e^{-8}$ |
| Weight decay | ✗ | ✗ |
| LR scheduler | Cosine | Cosine |
| Warmup ratio | 0.03 | 0.03 |
| Scale Bound | (0.5, 1.0) | (0.1, 10.0) |

Table A2: Detailed hyperparameters of various optimizers for C4 benchmark.

| Optimizer | $\beta_1$ | $\beta_2$ | $\beta_3$ | Eps. | 60M | 130M | 350M | 1B | 3B |
|---|---|---|---|---|---|---|---|---|---|
| APOLLO | 0.9 | 0.99 | − | 1e-6 | 2e-2 | 1e-2 | 1e-2 | 1e-2 | 1e-2 |
| Adabelief | 0.9 | 0.999 | − | 1e-12 | 1e-2 | 1e-2 | 1e-2 | 1e-3 | 1e-3 |
| Adafactor | 0.9 | − | − | − | 2e-3 | 2e-3 | 1e-3 | 5e-4 | 5e-4 |
| AdamW | 0.9 | 0.99 | − | 1e-8 | 3e-3 | 1e-3 | 1e-3 | 5e-4 | 5e-4 |
| Adam8bit | 0.9 | 0.99 | − | 1e-8 | 3e-3 | 1e-3 | 5e-4 | 5e-4 | 5e-4 |
| Adam-mini | 0.9 | 0.99 | − | 1e-8 | 3e-3 | 1e-3 | 5e-4 | 5e-4 | 1e-3 |
| Adamp | 0.9 | 0.98 | − | 1e-8 | 5e-3 | 1e-3 | 1e-3 | 5e-4 | 5e-4 |
| Adan | 0.9 | 0.92 | 0.99 | 1e-8 | 3e-3 | 3e-3 | 3e-3 | 1e-3 | 1e-3 |
| CAME | 0.9 | 0.98 | − | 1e-6 | 5e-3 | 2e-3 | 1e-3 | 5e-4 | 5e-4 |
| LAMB | 0.9 | 0.99 | − | 1e-6 | 5e-3 | 3e-3 | 1e-3 | 1e-3 | 1e-3 |
| Lion | 0.9 | 0.98 | − | − | 2e-4 | 2e-4 | 2e-4 | 1e-4 | 1e-4 |
| MARS+AdamW | 0.95 | 0.99 | − | 1e-8 | 5e-3 | 5e-3 | 5e-3 | 1e-3 | 1e-3 |
| MARS+Lion | 0.9 | 0.98 | − | 1e-8 | 5e-4 | 2e-4 | 2e-4 | 1e-4 | 1e-4 |
| MARS+Shampoo | 0.95 | 0.99 | − | 1e-8 | 5e-2 | 2e-2 | 1e-2 | 1e-2 | 1e-2 |
| Muon | 0.9 | 0.95 | − | 1e-8 | 4e-3 | 2e-3 | 1e-3 | 1e-3 | 1e-3 |
| Nadam | 0.9 | 0.99 | − | 1e-8 | 1e-3 | 1e-3 | 1e-3 | 5e-4 | 5e-4 |
| Prodigy | 0.9 | 0.95 | − | 1e-8 | 5e-1 | 1e-0 | 2e-0 | 2e-0 | 1e-0 |
| Radam | 0.9 | 0.99 | − | 1e-8 | 3e-3 | 1e-3 | 1e-3 | 5e-4 | 5e-4 |
| SOAP | 0.9 | 0.95 | − | 1e-8 | 3e-3 | 2e-3 | 1e-3 | 5e-4 | 5e-4 |
| Shampoo | 0.9 | 0.999 | − | 1e-8 | 5e-2 | 5e-2 | 2e-2 | 1e-2 | 1e-2 |
| Sophia | 0.9 | 0.99 | − | 1e-8 | 2e-4 | 2e-4 | 2e-4 | 1e-4 | 1e-4 |
| SGG+AdamW | 0.9 | 0.99 | 0.9 | 1e-8 | 3e-3 | 1e-3 | 1e-3 | 1e-3 | 1e-3 |
| SAC+Adam-mini | 0.9 | 0.99 | − | 1e-8 | 1e-2 | 5e-3 | 5e-3 | 5e-3 | 5e-3 |
| SAC+AdamW | 0.9 | 0.99 | − | 1e-8 | 1e-2 | 1e-2 | 1e-2 | 1e-2 | 1e-2 |
| SAC+Shampoo | 0.9 | 0.999 | − | 1e-8 | 5e-2 | 5e-2 | 3e-2 | 1e-2 | 1e-2 |

## A.2 LONG SEQUENCE PRE-TRAINING

We follow Gu & Dao (2023); Yang et al. (2025) for the standard long sequence modeling setups with 1.3B parameters. We adopt the 100B tokens sampled from the FineWeb-Edu dataset Penedo et al. (2024). The standard optimization setup is using the AdamW optimizer with a peak learning rate of 4e-4, weight decay of 0.1, and gradient clipping of 1.0. We employed the training setting with the tuned learning rate for different optimizers. A cosine annealing schedule with a 1B token warm-up period and a batch size of 0.5M tokens is used. All models employ the LLaMA 2 tokenizer with a vocabulary size of 32,000.

## A.3 IMAGE CLASSIFICATION

We follow BOCB benchmarks (Li et al., 2024b) for the fair comparison of popular optimizers on image classification tasks. We apply consistent setups for image classification tasks on CIFAR-100 (Krizhevsky et al., 2009) and ImageNet-1K (Krizhevsky et al., 2012) based on `OpenMixup` (Li et al., 2022b) codebase with 1 or 8 Nvidia A100 GPUs. Following the widely used modern training recipes, we consider three regular training settings for ImageNet-1K classification experiments for various backbones and optimizers, which could be transplanted to the proposed CIFAR-100 benchmarks. ResNet-50 is trained by the RSB-A2 setup for 300 epochs with advanced data augmentations in DeiT variants (Touvron et al., 2021), while DeiT-S is optimized with the standard DeiT training configuration. CIFAR-100 benchmarks adopt similar settings to the ImageNet-1K.

## A.4 LLM PEFT WITH COMMONSENSE REASONING TASKS

Following LLM-Adaptor (Hu et al., 2023), we evaluate eight Commonsense Reasoning tasks with top-1 accuracy (%) and GPU memory consumption, including BoolQ (Clark et al., 2019), PIQA (Bisk et al., 2020), SIQA (Sap et al., 2019), HellaSwag (Zellers et al., 2019), WinoGrande (Sakaguchi et al., 2021), ARC-Easy (ARC-E) and ARC-Challenge (ARC-C) (Clark et al., 2018), and OBQA (Mihaylov et al., 2018). As SFT setups in LLM-Adaptor, we combine the training datasets from all sub-tasks to fine-tune the pre-trained LLaMA-7B for 3 epochs using the AdamW optimizer with a basic learning rate of 1e-4, a batch size of 32, and the rank $r = 32$. Then, we evaluate each sub-task individually using its respective testing dataset. Three classical PEFT baselines, Prefix-tuning (Li & Liang, 2021),

Table A3: Details of the hyperparameters for the included optimizers and experiment settings. The `N/A` denotes that without this hyperparameter, the `Defaults` denotes other parameters set as defaults, M denotes the momentum, the params[1D] denotes the 1-dimensional parameters, $I^c$ and $n^c$ denotes cluster iteration and number of SGG, and SB denotes scale bound.

| Optimizer | Lr | Betas | Epsilon | Weight decay | Others params |
|---|---|---|---|---|---|
| AdamW | 2e-5 | (0.9, 0.999) | 1e-8 | N/A | Defaults |
| Adafactor | 2e-5 | (0.9, 0.999) | 1e-8 | N/A | Defaults |
| LAMB | 2e-4 | (0.9, 0.999) | 1e-8 | N/A | Defaults |
| RAdam | 2e-5 | (0.9, 0.999) | 1e-8 | N/A | Defaults |
| NAdam | 2e-5 | (0.9, 0.999) | 1e-8 | N/A | Defaults |
| Adan | 2e-5 | (0.98, 0.92, 0.99) | 1e-8 | N/A | Defaults |
| Shampoo | 2e-5 | (0.9, 0.999) | 1e-8 | N/A | Defaults |
| SOAP | 2e-5 | (0.9, 0.999) | 1e-8 | N/A | Defaults |
| Sophia | 2e-6 | (0.9, 0.999) | 1e-8 | N/A | Defaults |
| LION | 2e-6 | (0.9, 0.98) | 1e-8 | N/A | Defaults |
| CAME | 4e-5 | (0.9, 0.999) | 1e-8 | N/A | Defaults |
| SGD | 1e-5 | N/A | N/A | 1e-5 | M=0.99 |
| MARS+AdamW | 2e-5 | (0.9, 0.999) | 1e-8 | N/A | params[1D]: AdamW defaults |
| MARS+Shampoo | 1e-4 | (0.9, 0.999) | 1e-8 | N/A | params[1D]: Shampoo defaults |
| SGG+AdamW | 2e-5 | (0.9, 0.999) | 1e-8 | N/A | $I^c$=1k, $n^c$=5, beta3=0.9 |
| SGD+Adafactor | 2e-5 | (0.9, 0.999) | 1e-8 | N/A | $I^c$=1k, $n^c$=3, beta3=0.95 |
| SGD+Shampoo | 2e-5 | (0.9, 0.999) | 1e-8 | N/A | $I^c$=1k, $n^c$=5, beta3=0.9, SB=(0.5, 10.0) |
| SAC+AdamW | 2e-5 | (0.9, 0.999) | 1e-8 | N/A | SB=(0.5, 1.0) |
| SAC+Shampoo | 2e-5 | (0.9, 0.999) | 1e-8 | N/A | SB=(0.1, 10.0) |

Series Adapter (Series) (Houlsby et al., 2019), and Parallel Adapter (Parallel) (He et al., 2021), and three popular PEFT methods, DoRA (Liu et al., 2024d), GaLore (Zhao et al., 2024a), and Fira (Chen et al., 2024), are compared in Table 5. Our SAC consistently improves eight sub-tasks over LoRA without extra GPU memory, achieving competitive performances with well-designed PEFT methods with LoRA+SAC.

## A.5  MLLM SFT WITH LLAVA VARIANTS

To evaluate the generalization of the SAC-equipped optimizer, we conduct supervised fine-tuning on multiple LLaVA variants (Liu et al., 2024c)—i.e., LLaVA-v1.5-7B, LLaVA-LoRA, and LLaVA-v1.3—and compare against mainstream multimodal LLMs, e.g., BLIP (Li et al., 2022a), Instruct-BLIP (Dai et al., 2023), Qwen-VL (Bai et al., 2023), Qwen-VL-Chat, mPLUG-Owl2 (Ye et al., 2024), as well as LLaVA-family variants: Tiny-LLaVA (Zhou et al., 2024), MoE-LLaVA (Lin et al., 2024), LLaVA-Phi (Zhu et al., 2024b), LLaVA-NeXT (Liu et al., 2024b), LLaVA-MOD (Shu et al., 2024), and LLaVA-KD-2B (Cai et al., 2024). **Setup and settings.** Following LLaVA-v1.5, we use a pre-trained Vicuna-v1.5-7B (Chiang et al., 2023) as the language decoder. A pre-trained $2\times$MLP serves as the connector to align visual tokens to text tokens; it is trained for one epoch on the `LCS-558K` dataset. For the visual encoder, we adopt CLIP (Radford et al., 2021) to extract image representations. We validate three optimizers—AdamW, Adafactor, and LAMB—and additionally reproduce results for Muon (Jordan et al.) and MARS (Yuan et al., 2024). Optimizer hyperparameters and training details are summarized in Table A3.

**Supervised fine-tuning.** We freeze the visual encoder and update the connector and LLM parameters. For full-rank SFT, we use a learning rate of $2e-5$, a batch size of $64$, and train for one epoch on the `llava-v1.5-mix665k` dataset. To assess SAC under parameter- and memory-efficient regimes, we further evaluate Low-Rank Adaptation (LoRA) and 8-bit Quantized LoRA (Q-LoRA (Dettmers et al., 2024)). For both LoRA and Q-LoRA, we set the rank $r = 128$, scaling factor $\alpha = 256$, batch size 64, and train for one epoch; these configurations are based on LLaVA-v1.5.

---

**Algorithm 2** Modern LLM Optimizer (Inner Optimizer & Wrapper)

---

**Require:** Learning rate $\eta$, decay rates $\beta_1, \beta_2$, constant $\varepsilon$, weight decay $\lambda$, parameters $\theta$, gradients $G$
**Ensure:** Updated parameters $\theta$
1: **Initialize:** $m_0 \leftarrow 0, v_0 \leftarrow 0, t \leftarrow 0$
2: **Define groups:** $\mathcal{P}_{\text{hidden}}$ (2D matrices), $\mathcal{P}_{\text{other}}$ (scalars, vectors)

3: **function** MODERN LLM OPT$(\theta, G, \eta, \beta_1, \beta_2, \varepsilon, \lambda)$
4:     $t \leftarrow t + 1$          ▷ Timestep increment
    **Wrapper:** Gradient Adjustment
5:     **for** $\theta_i \in \theta$ **do**
6:         $G_i' \leftarrow G_i$          ▷ Initialize with raw gradient
7:         **if** $\theta_i \in \mathcal{P}_{\text{hidden}}$ **then**
8:             $G_i' \leftarrow \text{RefineGradient}(G_i, \theta_i, \eta)$      ▷ Apply gradient refinement for 2D weights
9:         **end if**
    **Inner Optimizer:** Parameter Update (*e.g.,* AdamW)
10:         $m_i \leftarrow \beta_1 m_i + (1 - \beta_1)G_i'$
11:         $v_i \leftarrow \beta_2 v_i + (1 - \beta_2)(G_i')^2$
12:         $\hat{m}_i \leftarrow m_i/(1 - \beta_1^t)$
13:         $\hat{v}_i \leftarrow v_i/(1 - \beta_2^t)$
14:         $\theta_i \leftarrow \theta_i - \eta \cdot \hat{m}_i/(\sqrt{\hat{v}_i} + \varepsilon) - \eta \cdot \lambda \cdot \theta_i$
15:     **end for**
16:     **return** $\theta$          ▷ Return updated parameters
17: **end function**

---

