# OpenReview forum: "SAC: Adaptive Learning Rate Scaling with Architectural Constraints"
_ICLR.cc/2026/Conference — Submitted to ICLR 2026_

### Official Review · Reviewer_FoCz · 2025-10-30

**Soundness:** 3
**Presentation:** 2
**Contribution:** 2
**Rating:** 4
**Confidence:** 3

**Summary:**

The paper introduces SAC (Scaling with Architectural Constraints), a light‑weight wrapper around Adam‑family optimizers that multiplies the usual Adam update by an architecture‑aware scale factor $S_t$. The factor is computed online from robust gradient statistics at several structural granularities (block, layer, and optionally group/parameter), so that the effective per‑parameter learning rate becomes
$
\alpha_\theta = \eta\cdot c_l \cdot s_b \cdot r_\theta,\qquad \theta\in\mathcal P_{l,b},
$
with a layer factor $c_l$ that equalizes update magnitudes across depth based on a ratio of MAD dispersions, optionally EMA‑smoothed, a block factor $s_b$ that redistributes learning‑rate “budget” across blocks within a layer via a temperatured softmax, and an optional within‑block term $r_\theta$. The combined factor is applied multiplicatively to the base Adam‑like step.

**Strengths:**

1. The Optimizer Design Plane surfaces the Coordinated Optimization Zone and motivates adding spatial structure to Adam‑style methods.
2. Using MAD for dispersion plus a budgeted softmax to keep a per‑layer LR budget is principled and effective; ablations show MAD’s advantage.
3. The pipeline and partition are easy to follow.

**Weaknesses:**

1. Eq. (7) defines $\phi_{l,b}$ via log‑RMS, whereas Algorithm 1 computes a MAD‑based log‑ratio; the paper doesn’t state which variant produced Tables 1–6.
2. The text says SAC adds “negligible overhead,” but Table 1 shows sizable increases in optimizer‑step time for SAC+AdamW (e.g., 350M: 0.0045 s → 0.0401 s; 1B: 0.0762 s → 0.1089 s). It would be better to provide end‑to‑end wall‑clock (including forward/backward) and clarify communication costs for global/layer statistics.
3. Core knobs $\gamma,\beta,\rho$ and clamp bounds $[S_{\min},S_{\max}]$ are not specified per experiment in Appendix; defaults and sensitivity are essential.
4. Many gains are modest; results lack seeds/variance/CIs across Tables 1–6.
5. The text says SAC “matches or surpasses PEFT baselines,” yet Table 5 shows DoRA has a higher average than LoRA+SAC. Either add SAC+DoRA or soften the claim.
6. (i) The optional $r_\theta$ is introduced but not implemented in Algorithm 1. (ii) SAC scales the data‑driven step but not decoupled weight decay (Algorithm 1), thereby changing their relative strengths.
7.  Appendix Table A2 uses different LR grids per optimizer (e.g., SAC+AdamW fixed at (10^{-2}) across sizes, while AdamW baselines use size‑dependent smaller peaks). Ensure identical sweep budgets and report best‑of‑sweep for each optimizer with seeds.

**Questions:**

1. Which definition was used in Tables 1–6. Eq. (7) (log‑RMS) or Algorithm 1’s MAD‑based score? If both were tried, how do results differ? Please align equation, pseudocode, and code.
2. What exact values were used for $\gamma,\beta,\rho,S_{\min},S_{\max}$ in each table (LLM, vision, MLLM)? Can you add sensitivity plots and recommended defaults?
3. Provide end‑to‑end wall‑clock per step and fraction due to reductions for global/layer MADs; how does overhead scale with data/tensor parallel sizes?
4. Did you try scaling decoupled weight decay by $c_l s_{l,b}$? If so, how did this affect stability/convergence vs. the current choice (Algorithm 1)?
5. How does clipping (used in long‑sequence setups) interact with $c_l$ (which inflates when dispersion is small)? Any clamping or pre‑clip computation?
6. Any experiments with head/row/column‑wise $r_\theta$? If not used, please clarify and update Eq. (3) to match the implemented variant.
7. For Table 1 and Table 4, could you report best‑of‑equal‑budget LR sweeps (same grid, same seed counts) and include error bars?
8. Can you evaluate SAC+DoRA or SAC+Fira to support the statement “matches or surpasses PEFT baselines”?

---

### Official Review · Reviewer_WE76 · 2025-10-31

**Soundness:** 2
**Presentation:** 2
**Contribution:** 2
**Rating:** 2
**Confidence:** 3

**Summary:**

The paper introduces SAC (Scaling with Architectural Constraints) — a wrapper around adaptive optimizers (e.g., Adam/AdamW) that computes hierarchical, architecture-aware scaling factors for parameter updates. SAC aims to combine temporal smoothing (as in Adam) with spatial coordination (across layers and blocks), using robust gradient statistics to dynamically adjust learning rate multipliers at multiple levels (block/layer/global). The method is simple, computationally light (as experiments demonstrate), and shows empirical improvements across LLM pre-training, vision tasks.

**Strengths:**

__1) Motivated idea__
The idea of adding “architecturally coordinated optimization” to adaptive optimizers is understandable and eliminates their real limitations, which ignore structural correlations between parameters.

__2) Design__
SAC is easy to implement into existing optimizers. This makes it possible to use this framework in practice.

__3) Comprehensive experiments__
The paper includes evaluations across multiple domains (language, vision, PEFT) and performs several ablations.

**Weaknesses:**

__1) Limited novelty and conceptual depth.__
Despite the fact that the method is well developed, its conceptual novelty is weaker than stated. Adaptive architecture-based or layer-based scaling was investigated earlier. Combining these concepts by adding a structured scaling factor to the step looks heuristic. From my point of view, SAC can be viewed more as a pragmatic simplification than as a fundamentally new paradigm.

__2) Lack of theoretical grounding.__
The article does not provide an analysis of convergence or stability, as well as a theoretical justification for using MAD-based scaling. The choice of parameters (logarithmic average, the ratio of the layers $\frac{\delta_{global}}{\delta_{layer}}$) is heuristic. Without even a simple proof sketch or binding, statements about “coordinated optimization zones” remain more conceptual than formal.

__3) Experimental problems.__

a) The authors do not provide any statistical studies of the results obtained. This greatly reduces the significance of their results.

b) The improvement of the SAC wrapper over the baseline optimizers provides an improvement, however insignificant (0.2-1%). Coupled with the fact that the authors do not provide statistical studies, it is impossible to say with certainty that SAC improves the basic optimizers.

c) In experiments with large language models, the authors validate the methods on models with a maximum of 3B parameters. It would be great to see the robustness of their method to more stable models of a larger size with more than 7B parameters.

__4) Concerns of adapting across architectures.__
SAC depends on multiple ad hoc components: MAD statistics, logarithmic rescaling, clamping $[S_{min}, S_{max}]$, and hand-defined grouping of layers/blocks. This raises concerns about generality and reproducibility across architectures (e.g., hybrids, MoE models).

**Questions:**

__Overall Assessment__

SAC is a well-implemented wrapper for basic adaptive optimizers. However, its conceptual and theoretical novelty is limited, and the paper currently lacks sufficient rigor to justify claims of a new optimization paradigm. Although the experimental section looks fundamental, it has its drawbacks, such as the lack of statistical significance of improvements to the proposed method.

---

### Official Review · Reviewer_TB2v · 2025-11-01

**Soundness:** 4
**Presentation:** 3
**Contribution:** 4
**Rating:** 8
**Confidence:** 4

**Summary:**

This paper introduces **SAC (Scaling with Architectural Constraints)** — an optimizer *wrapper* that enhances adaptive optimizers (e.g., AdamW, Adam-mini, Shampoo) by **modulating per-parameter learning rates through lightweight, architecture-aware constraints**.

The key insight is that modern LLM optimization is dominated by **temporal smoothing** (historical gradients) but remains blind to **spatial structuring** (architectural hierarchy). SAC addresses this by introducing hierarchical scale factors — at parameter, group, layer, and block levels — to coordinate updates within the model’s structure, without adding significant computational cost.

Extensive experiments cover:
- **LLM pretraining (C4 dataset, LLaMA models 60M–3B)**
- **Vision classification (CIFAR-100, ImageNet-1K)**
- **Long-sequence modeling (Gated DeltaNet 1.3B)**
- **PEFT and multimodal fine-tuning (LLaVA, LoRA, etc.)**

Empirically, SAC+AdamW achieves state-of-the-art perplexity with minimal overhead, outperforming strong baselines like MARS, Muon, and Shampoo by up to **30% improvement in convergence or accuracy metrics**.

**Strengths:**

1. **Strong conceptual novelty** — identifies and operationalizes the underexplored “spatial structuring” dimension in optimizers.
2. **Extensive empirical validation** — over 20 benchmarks across NLP, vision, and multimodal tasks, with consistent performance gains.
3. **High practical value** — easily implementable, compatible with standard frameworks (PyTorch), and adds minimal overhead.
4. **Robustness and interpretability** — hierarchical factors (layer, block, group) make optimization more interpretable, as seen in Figure 3’s LR distributions.
5. **Cross-domain generality** — SAC’s success in both LLMs and CNN/Transformer models is uncommon for optimizer work.
6. **Comprehensive ablation and comparison** — with state-of-the-art optimizers like Muon, MARS, and APOLLO.

**Weaknesses:**

1. **Lack of theoretical grounding:**
   While empirical results are strong, there is no formal proof or convergence bound. The method’s stability under gradient noise or distributed variance is unaddressed.

2. **Over-reliance on empirical heuristics:**
   Hyperparameter selection (γ, β, ρ) is justified qualitatively but not theoretically. An analysis of sensitivity or scaling behavior would strengthen the argument.

3. **Limited interpretability of scaling dynamics:**
   Although Figure 3 shows per-block LR distributions, it remains unclear *why* specific architectural hierarchies (e.g., block-wise vs. layer-wise) improve performance.

4. **Missing discussion on training cost vs. accuracy trade-offs at ultra-large scale (≥7B):**
   Appendix A hints at robustness, but empirical validation stops at 3B.

5. **Minor editorial issues:**
   Some redundancy between main and appendix results; writing could be more streamlined in methodology sections.

**Questions:**

1. How sensitive is SAC to architectural partitioning granularity? Would arbitrary grouping (not aligned to model structure) still yield improvements?
2. Could the hierarchical scale factors be *learned dynamically* (e.g., via meta-learning) instead of hand-designed MAD statistics?
3. What are the convergence characteristics under high learning rate regimes or mixed-precision instability?
4. How does SAC interact with low-rank fine-tuning (e.g., LoRA/Q-LoRA) in terms of memory scaling?
5. For distributed training, does aggregation of per-layer statistics introduce synchronization overhead at very large scales (100B+ params)?
6. Can the authors provide theoretical justification (even approximate) for the observed stability improvements?

---

### Official Review · Reviewer_NwGh · 2025-11-01

**Soundness:** 3
**Presentation:** 3
**Contribution:** 3
**Rating:** 4
**Confidence:** 4

**Summary:**

The paper introduces SAC (Scaling with Architectural Constraints), a lightweight wrapper placed on top of optimzier. SAC decomposes the effective learning rate into hierarchical factors at block/layer/group/parameter levels and multiplies these with the base per-parameter rate to coordinate updates across the model.

**Strengths:**

* **Clarity and thoroughness**： The methodology is clearly articulated，and easy to understand.
* **Scoping that fits distributed training**: The partition/indexing scheme is claimed to align with TP/FSDP and add only a small state and constant-time index lookups. This is important, as otherwise the added techniques is not general due to very complex design.
* **drop-in idea with imporvement**: Clean wrapper with some performance gain at the cost of overhead.

**Weaknesses:**

* Theory gap (conditioning & mechanisms): The paper’s own “Limitations” admits that rigorous theoretical analysis would help clarify mechanisms, especially effects on model conditioning. I feel the current methods are too heuristic with too many hyperparameters.
* Sensitivity/generalization knobs: The approach introduces several new hyperparameters, lacking a guide for how to select them

**Questions:**

* The method introduces many hyperparameters; how to choose them, especially without a theory-driven approach?
* The optimizer introduces extra overhead, actually. Do you expect a bigger/smaller overhead in the FSDP/TP training setting?
* Can the method be extended to Zero-2/3 with model parallelism? If so, what is the overhead?

---

### Meta-Review · Area_Chair_QkWg · 2026-01-05

**Summary:**

The paper propose SAC, which is a new optimizer wrapper that change learning rates based on model architecture layers and blocks. It try to balance performance and memory by use hierarchical factors to coordinate how parameters is updated during training.

The reviewers has raised several critical concern that make acceptance difficult at this stage, including

1. there is a big lack of theoretical grounding; most reviewers agree there is no formal proof for convergence or why the MAD-based scaling actually works.
2. The computational overhead is worrying because the optimizer step time increase significantly in Table 1, which contradict the authors' claim of "negligible" cost. Furthermore, the method introduce many new hyperparameters like $\gamma, \beta$, and $\rho$, but the paper does not give clear guide on how to choose them for different models.
3. Reviewers also point out experimental issues, such as the lack of statistical error bars and the fact that tests only go up to 3B parameters, leaving the performance on larger models unknown
4. there is some inconsistency between the formulas in the text and the pseudocode in Algorithm 1 regarding which statistics is used

**Reviewer Concerns:**

NA - the authors has not submitted rebuttal

**Reviewer Scores:**

NA -  the authors has not submitted rebuttal. Most of the reviewers agreed that the paper is not ready for acceptance

---

### Decision · Program_Chairs · 2026-01-26

Reject